# T Cell Immunosenescence in Aging, Obesity, and Cardiovascular Disease

**DOI:** 10.3390/cells10092435

**Published:** 2021-09-15

**Authors:** Kohsuke Shirakawa, Motoaki Sano

**Affiliations:** 1Department of Cardiovascular Medicine, Graduate School of Medicine, Juntendo University, Bunkyo-ku, Tokyo 1138421, Japan; shirakawa19840905@gmail.com; 2Department of Cardiology, Keio University School of Medicine, Shinjuku-ku, Tokyo 1608582, Japan

**Keywords:** obesity, cardiovascular disease, immunosenescence, T cell

## Abstract

Although advances in preventive medicine have greatly improved prognosis, cardiovascular disease (CVD) remains the leading cause of death worldwide. This clearly indicates that there remain residual cardiovascular risks that have not been targeted by conventional therapies. The results of multiple animal studies and clinical trials clearly indicate that inflammation is the most important residual risk and a potential target for CVD prevention. The immune cell network is intricately regulated to maintain homeostasis. Ageing associated changes to the immune system occurs in both innate and adaptive immune cells, however T cells are most susceptible to this process. T-cell changes due to thymic degeneration and homeostatic proliferation, metabolic abnormalities, telomere length shortening, and epigenetic changes associated with aging and obesity may not only reduce normal immune function, but also induce inflammatory tendencies, a process referred to as immunosenescence. Since the disruption of biological homeostasis by T cell immunosenescence is closely related to the development and progression of CVD via inflammation, senescent T cells are attracting attention as a new therapeutic target. In this review, we discuss the relationship between CVD and T cell immunosenescence associated with aging and obesity.

## 1. Introduction

Despite the identification of risk factors that promote atherosclerosis, including hypertension, dyslipidemia, hyperglycemia, and smoking, as well as the development of therapies to ameliorate each risk factor, cardiovascular disease (CVD) remains the leading cause of death worldwide. In particular, targeting of inflammatory processes in experimental animal models has been demonstrated to be beneficial in ameliorating myocardial injury and promoting healing [1,2,3]. Recent clinical trials have confirmed that inflammation is not only a residual risk factor, but also an effective therapeutic target for secondary prevention of CVD [4,5].

It has been reported that a certain percentage of hematopoietic stem/progenitor cells with a specific genetic mutation called clonal hematopoiesis of indeterminate potential (CHIP) appear in the elderly, which is not only associated with the risk of hematopoietic tumors but also with the development of atherosclerotic cardiovascular disease [6]. In addition, environmental stresses such as lack of exercise and sleep disturbance have been shown to accelerate the progression of atherosclerosis by increasing the production of inflammatory monocytes and neutrophils in the bone marrow [7,8].

Besides, cellular senescence has been implicated as a major cause in age-related functional decline and chronic low-grade inflammation observed during aging [9,10]. It has several triggers, including DNA damage, telomere length shortening, metabolic dysfunction, and organelle stress, which occur in multiple cell types [11,12]. Senescent cells secrete pro-inflammatory factors known as the senescence-associated secretory phenotype (SASP). Senescent cells accumulate in organisms with aging and create chronic inflammation in the surrounding tissues via SASP, which causes a functional decline of organs and contributes to the development of multiple age-related chronic diseases, including CVD, endocrine and metabolic diseases, cancer, Alzheimer’s disease, and autoimmune diseases [4,5,9,10,11,13]. Various types of cells acquire a senescent phenotype during chronological aging. Age-related changes in the immune system play an important role in the development of various age-related diseases [14,15,16]. 

The immune system is one of the most important regulators of physiological homeostasis. With aging, immune function shows characteristic changes, including progressive decline in acquired immune response to foreign pathogens (e.g., decreased vaccine efficiency) and a tendency to produce excessive inflammatory responses [17,18]. This phenomenon is referred to as immune senescence or immunosenescence [17,18] and is thought to be closely related to the development of CVD because the SASP factor secreted by senescent immune cells causes progressive organ remodeling [15,16]. Age-related changes occur in most types of cells in both innate and adaptive immune systems, however generation and homeostasis of T cells are particularly susceptible to ageing due to the involution of the thymus [19].

Visceral obesity is associated with CVD, heart failure (HF), diabetes, non-alcoholic fatty liver disease, and nephropathy, resulting in increased mortality [20,21]. Chronic low-grade inflammation in visceral adipose tissue (VAT) contributes to the development of obesity-associated comorbidities [22]. This inflammation involves a complicated network of responses of the immune system, including acquired immunity (T and B cells) and innate immunity, such as macrophages [23,24]. We and other investigators have previously demonstrated that obesity accelerates T cell immunosenescence, independent of chronological aging [25,26]. A unique population of CD^4+^ T cells that constitutively express programmed cell death 1 (PD-1) and CD153 preferentially increase and accumulate in the VAT of HFD-fed obese mice; their features are highly reminiscent of senescence-associated T (SA-T) cells that accumulate in secondary lymphoid tissue with aging [25]. They produce large amounts of proinflammatory cytokines, such as osteopontin (OPN), leading to chronic inflammation of the VAT and the subsequent development of insulin resistance [25,27,28].

Thus, obesity and aging may independently induce immune senescence and accelerate the pathological progression of CVD through the acquisition of SASP by senescent immune cells. In this review, we summarize the current understanding of immune senescence and explore the evidence linking aging, obesity, and T cell immunosenescence with CVD.

## 2. Immunosenescence as a Function

### 2.1. Immunosenescence Changes with Aging

Age-related changes in the immune system play an important role in the pathogenesis and etiology of multiple age-related diseases [14]. The immune system has a decreased capacity for acquired immune responses and an increased pro-inflammatory predisposition and autoimmune response with aging [29]. In fact, elderly people have diminished immune function, including susceptibility to infection, reduced vaccine efficacy, and reactivation of latent viral infections, resulting in a diminished normal immune response that protects the body from infection and eliminates pathogens. Thus, aging-associated remodeling of the immune system contributes to the development of systemic inflammation and autoimmune diseases, and increases morbidity and mortality in the elderly, referred to as immunosenescence [18]. Injured tissues have the ability to converge inflammation in order to maintain organ homeostasis; however, they cannot completely return to their original state and thus pathological remodeling of the organ occurs, including deposition of extracellular matrix (ECM). When inflammation is prolonged due to immunosenescence, the negative legacy (increased entropy) accumulated in the tissues causes organ dysfunction. 

Age-related quantitative or qualitative changes can occur in all immune cell lines [30]. During the chronological aging process, the most remarkable changes are observed in the adaptive immune system, particularly in T cells. Surprisingly, the number of peripheral T cells is sustained with age, despite a marked decrease in T cell production due to thymic involution beginning early in human life [31]. Since naïve T cells in the periphery are primarily maintained by homeostatic proliferation after thymic involution, there is no doubt that T cells are strongly affected by aging, unlike other immune cell types that depend on the production of hematopoietic stem cells (HSCs) [31,32].

### 2.2. Thymic Involution with Aging

The thymus is the primary lymphoid organ for generating self-restricted and self-tolerant functional T cells; their precursors undergo positive selection and negative selection in the thymic cortex and medulla, respectively. Thymic involution is caused by decreased stem cell activity of medullary thymic epithelial cells, which are the primary thymic stromal cells in the production of T cells [33,34]. This decrease in stem-cell activity has been suggested to depend on the sustainable production of active T cells immediately after birth, rather than on the effects of aging [35]. 

### 2.3. Homeostatic Proliferation with Aging

Despite the progressive decline in T cell output with thymic involution, homeostatic proliferation complements the maintenance of absolute T cell numbers, so that the overall peripheral T cell population is largely maintained [32,36,37]. In fact, the proliferation of T cells is dominant in the blood of older individuals [38]. The fact that naïve T cells in the periphery are maintained by the proliferation of T cells produced in early postnatal life is completely different from that of other immune cell types, and this process contributes to immunosenescence.

During homeostatic proliferation, naïve T cells are maintained in the periphery by weak T-cell receptor (TCR) signaling from the self-peptide-major histocompatibility complex (MHC) expressed by stromal cells in secondary lymphoid tissues, and by signaling from cytokines important for lymphocyte maintenance, such as interleukin-7 and interleukin-15 [39]. 

T cells that receive stronger TCR signals are preferentially expanded into larger clones that occupy an increasing proportion of the pool of T cells [29,40,41,42]. In humans, there is little evidence of any decrease in the diversity of T cells in the elderly [38,41]; however, it has been reported that the diversity of the TCR repertoire on naïve CD^4+^ and CD^8+^ T cells is markedly lower in the elderly than in young adults [42]. Furthermore, homeostatic proliferation drives a rapid decline in telomere length [43], with telomere shortening contributing to a persistent DNA damage response during replicative senescence [44]. Thus, homeostatic proliferation leads to deflection and narrowing of the repertoire and efficient amplification of self-responsive clones, which may contribute to decreased acquired immune response capacity and increased predisposition to self-response.

### 2.4. Phenotypic Changes in T Cells with Aging

Senescent T cells lose quiescence and share many features with differentiated T cells [41]. As mentioned previously, homeostatic proliferation drives telomere length shortening [43], which contributes to a persistent DNA damage response during replicative senescence [44]. CD^4+^ naïve T cells were found to have telomeres 1.4 ± 0.1 kb longer than the telomeres of memory cells from the same donor [45]. The composition of peripheral T cell subsets changes with age. The T cell population shows skewing from naïve (CD62L^high^ CD44^low^) to memory (CD62L^low^ CD44^high^) phenotype with aging [36,37]. The regulation of memory T cell populations is complicated, with an important distinction between the antigen-dependent and-independent mechanisms of memory T cell maintenance [41]. As a result of homeostatic proliferation, virtual memory T cells, which express memory-type cell surface markers and have enhanced effector functions despite not being stimulated by foreign antigens, have been found to differentiate and be classified as memory T cells [36,46,47]. 

Furthermore, recent single-cell RNA sequencing has shown that aging increases cell-to-cell transcriptional variability and heterogeneity in stimulated murine CD^4+^ T cells [48]. Thus, homeostatic proliferation is thought to lead to the constriction and bias of the peripheral T cell repertoire and self-responsive clones, which contributes to the development of T cell senescence [49]. 

### 2.5. T Cell Subset Found to Increase with Age in Mice

In terms of memory phenotypes, a unique population expressing programmed cell death 1 (PD-1), a negative costimulatory receptor for TCR signaling, and CD153, a TNF superfamily protein, increases with age in mice [27]. CD153+PD-1+CD44hiCD^4+^ T cells show signatures of cellular senescence, including a marked increase in senescence-related gene expression and nuclear heterochromatin foci, and compromised proliferation and production of regular cytokines upon TCR stimulation. The expression of miR-181a decreases with age, especially in CD153+PD-1+CD44hiCD^4+^ T cells in mice [27]. A decrease in miR-181a expression impairs TCR sensitivity by increasing the activity of dual specific phosphatase 6 [50]. Interestingly, these CD^4+^ T cells secrete large amounts of OPN at the cost of normal T cell function. OPN is a matricellular protein that mediates diverse biological functions [51,52,53]. It functions as a proinflammatory cytokine and promotes cell-mediated immune responses [54,55]. Thus, CD153+PD-1+CD44hiCD^4+^ T cells are defined as SA-T cells [27]. SA-T cells increase rapidly in the germinal center of secondary lymphoid tissue, not only with age, but also with leukemia and systemic autoimmune diseases; they are involved in acquired immune dysfunction and chronic tissue inflammation associated with these conditions [56].

## 3. Obesity and T Cell Senescence

### 3.1. Visceral Obesity and Immune Function

Visceral obesity is associated with insulin resistance, diabetes, fatty liver disease, CVD, and nephropathy, as well as reduced life expectancy [20,57]. Obesity-associated VAT inflammation involves a complex network of immune cells, including types involved in the acquired immune response (multiple subsets of T cells and B cells), innate immune cells such as macrophages [23,24,58], and chronic low-grade inflammation in VAT linked to the development of obesity-associated comorbidities. In cases of severe obesity, VAT can constitute up to 50–60% of the total body mass, containing millions of leukocytes per gram of adipose tissue [59]. Normal immune function does not occur in chronic inflammation [60,61].

### 3.2. Visceral Obesity and Cellular Senescence 

Cellular senescence is a fate characterized by irreversible cell cycle arrest and acquisition of SASP [62]. It occurs in response to several triggers, including DNA damage, telomere dysfunction, inflammation, metabolic dysfunction, and epigenetic changes [10,63].

Senescent cells accumulate during obesity [64,65,66] and drive the development of multiple obesity-related diseases, including CVD, pulmonary fibrosis, neurodegeneration, and osteoporosis [67,68,69]. Senescent cells expressing SA β-gal activity and p53 levels increase with obesity [64]. Furthermore, the clearance of senescent cells can ameliorate pathology in mice predisposed to obesity-related diseases [68,70,71]. 

Obesity promotes cellular aging via multiple mechanisms. Obesity induces oxidative stress and inflammation, which causes telomere shortening [72]. Telomere length is inversely associated with body mass index (BMI) [73]. Higher adiposity is associated with shorter telomere length, and telomere length is inversely correlated with serum concentration of leptin [74,75]. Obesity triggers widespread changes in gene expression in multiple organs [76] and methylation changes in blood leukocyte DNA, which can lead to immune dysfunction [77,78]. 

### 3.3. T Cell Senescence in VAT

CD^4+^ T cells are recognized as a central controller of the chronic adipose tissue inflammation associated with visceral obesity [79,80,81,82]. Numbers of VAT CD^4+^ T cells increase as adipose tissue expands in obesity. Visceral obesity increases the production of proinflammatory mediators from CD^4+^ T cells in the adipose tissue. IFN-γ-producing T-bet^+^ CD^4+^ T cells enhance adipose tissue inflammation via pro-inflammatory M1 macrophage activation [79,80,81,82]. Senescent cells accumulate in visceral fat in obesity [64,66] and contribute to the development of VAT inflammation. Depletion of senescent cells by treatment with senolytic agents has been shown to ameliorate adipose inflammation and metabolic dysfunction in obese mice [68]. 

### 3.4. Senescence-Associated CD^4+^ T Cells in VAT with Obesity

We discovered that diet-induced obesity reduced the frequency of CD44^lo^ CD62L^hi^ naïve CD^4+^ T cells and increased the frequency of both CD44^hi^ CD62L^lo^ memory phenotype CD^4+^ T cells in VAT and a unique population of CD44^hi^ CD62L^lo^ CD^4+^ T cells that constitutively express PD-1 and CD153 exhibit cellular senescence [25]. They express the senescence markers SA β-gal, γ-H2AX, and *Cdkn1a*, and cause VAT inflammation by producing large amounts of OPN at the cost of normal T cell functions [25]. Such a qualitative change in CD^4+^ T cells is similar to that in SA-T cells observed during chronological aging.

CD153^+^ PD-1^+^ CD^4+^ T cells play a central role in the pathogenesis of chronic inflammation in visceral adipose tissue through the production of OPN (34). Specifically, they stimulate the production of IFNγ from CD^8+^ T cells and Th1 cells to induce macrophage bias toward M1, promote migration of pro-inflammatory macrophages, and activate the production of IL-17 by Th17 and pathogenic antibodies by B cells, while suppressing the production of the anti-inflammatory cytokine IL-10 by Treg and B cells. The VAT of high-fat diet (HFD)-induced obese mice vaccinated with CD153-CpG showed a significant decrease in adipose senescent T cell accumulation and concomitant improvement in glucose tolerance and insulin resistance [83].

The antigens presented by major histocompatibility complex (MHC) class II proteins may induce the expansion of CD^4+^ T cells and their differentiation into inflammatory effectors in VAT during obesity, although the nature of these antigens is not yet clear [79]. In high-fat diet-fed μMT mice lacking mature B cells, the numbers and proportions of CD44^hi^CD62L^lo^CD^4+^ T cells in VAT were markedly reduced; this effect was associated with a decrease in CD153^+^ PD-1^+^ CD^4+^ T cells [25,27]. These results indicate that B cells may function as antigen-presenting cells that promote the generation of senescent CD^4+^ T cells in VAT.

### 3.5. Effect of Weight Loss on CVD and T Cell Senescence 

Weight loss improves glycemic control and reduces CVD risk, but epidemiological studies have shown conflicting results regarding its effect on cardiovascular events and longevity in obese people with type 2 diabetes [84,85,86]. For example, a prospective randomized study demonstrated that weight loss did not contribute to reduced cardiovascular morbidity and mortality in overweight or obese adults with type 2 diabetes at a median follow-up of almost 10 years [84]. Notably, we have shown that CD153^+^PD-1^+^CD44^hi^CD^4+^T cells in VAT are long-lived and not easily eliminated as a negative legacy of obesity after weight loss [87]. VAT CD153^+^PD-1^+^CD44^hi^CD^4+^T cells maintain a self-sustaining chronic inflammatory loop via the production of large amounts of OPN, which may maintain continuous cardiac inflammation during obesity.

### 3.6. Thymic Involution and Homeostatic Proliferation with Obesity

Diet-induced obesity accelerates thymic involution and age-related reduction in T cell maturation. TCR spectratyping has shown that DIO reduces thymopoiesis and restricts TCR diversity; moreover, obesity reduces thymic output in middle-aged humans independent of type 2 diabetes [88]. Furthermore, VAT T cells markedly restricted TCR diversity compared to splenic T cells, which was further compromised by obesity [86]. In humans, the homeostatic proliferation of both circulating CD^4+^ and CD^8+^ T cells is accelerated in obese individuals [89]. These results indicate that extensive homeostatic proliferation in obese individuals may lead to the emergence of dysfunctional T cells with the features of senescent cells in the VAT.

### 3.7. Metabolic Dysfunction and Continuous Antigen Stimulation with Obesity

Obese patients generally exhibit higher concentrations of leptin and insulin than nonobese controls [90]. Leptin, an adipokine secreted primarily by adipocytes, activates T cells and promotes their differentiation into the IFN-γ-producing Th1 phenotype [91,92]. Insulin also modulates T cell proliferation and IFN-γ production by controlling cell metabolism [93]. MHC class II deficiency ameliorated HFD-induced VAT inflammation as well as insulin resistance [83,86,94]. This suggests that antigens (probably self-peptides) presented by MHC class II induce the expansion of CD^4+^ T cells in VAT and their differentiation into inflammatory effectors during the development of HFD-induced obesity. Repeated antigen stimulation induces telomere shortening [95]. Accordingly, continuous antigens, possible self-antigens, and stimulation may contribute to T cell senescence in VAT.

### 3.8. Obesity and B Cell Senescence

Visceral adiposity also impairs B cell function and induces a pro-inflammatory phenotype [96]. B cells in the VAT express higher levels of proinflammatory markers, such as NF-kB, than do those in the spleen, which promote insulin resistance through the production of pathogenic IgG [97]. Plasma IgG levels were elevated in HFD mice with visceral fat obesity over those in mice that consumed a normal diet [25]. Serum IgG autoantibodies are increased in obese individuals with insulin resistance over those in individuals with insulin sensitivity [97]. Interestingly, leptin has been reported to increase the production of autoimmune antibodies [98]. It has been suggested that B cells play an important role in the maturation of aging T cells in VAT as antigen-presenting cells [25].

Thus, obesity induces a reduction in thymic output, extensive homeostatic proliferation, and repeated antigen stimulation, all of which likely contribute to T cell senescence (Figure 1).

## 4. Immunosenescence in Cardiovascular Diseases 

Age-related thymic regression and continuous antigen stimulation cause T cell subsets to change from a naïve to a memory phenotype [99], accompanied by downregulation of co-stimulatory molecules such as CD27 and CD28 [100]. These changes are considered to be a hallmark of human T cell senescence and are associated with increased susceptibility to age-related diseases such as infections, autoimmune diseases, cancer, and CVD [101]. 

### 4.1. Senescence of Human CD^4+^ T Cells

Shortened leukocyte telomeres have been observed in patients with coronary artery disease (CAD) and have been associated with the risk of myocardial infarction, independent of other common risk factors [102,103]. IFN-γ-producing CD^4+^ and CD^8+^ T cells increased in the peripheral blood of patients with acute coronary syndrome; the T cell chronotype of patients with unstable angina used a similar array of antigen receptors [104]. CD^4+^CD28^null^T cells recognize specific antigens and infiltrate into unstable coronary plaques via clonal expansion [105,106,107]. These results suggest that persistent stimulation by certain antigens may promote T cell senescence, and that their proinflammatory phenotype may exacerbate CVD.

CD28^null^CD^4+^ T cells secrete large amounts of interferon-gamma (IFN-γ) [107]. These T cells accumulate in the heart-draining lymph nodes of aged mice; furthermore, adoptive transfer of these causes a proinflammatory response in young mice [108]. Interestingly, when CD^4+^ T cells isolated from the peripheral blood of healthy humans were transplanted into humanized young lymphocyte-deficient mice, transplanted naïve CD^4+^ T cells (defined as CCR7^+^CD45RO^−^) underwent homeostatic expansion, upregulated expression of PD-1, and strongly shifted towards effector/memory (CCR7^−^CD45RO^+^) and terminally differentiated (CCR7^−^CD45RO^−^) phenotypes, as typically seen in the elderly. Expanded senescent CD^4+^ T cells infiltrate the heart and promote myocardial inflammation and stress response, followed by age-related cardiac dysfunction [109]. This result demonstrated that T cell senescence by homeostatic proliferation is directly involved in age-associated cardiac dysfunction.

Recently, human T cell senescence has been shown to be actively maintained by p38 activation [110]. During T-cell senescence, the TAK1-binding protein 1 (TAB1)-AMP-activated protein kinase (AMPK) complex senses low glucose concentration and DNA damage, resulting in sustained activation of downstream p38. The resulting reduction in proliferative capacity and telomerase activity by this mechanism can be restored by blocking p38 [111]. In addition, the inhibition of stress-induced sestrins, which are markedly increased in CD27^−^CD28^−^CD^4+^ T cells and activate Erk-Jnk-p38 mitogen-activated protein kinases, was shown to restore antigen-specific responses and responses to vaccines [112].

### 4.2. Senescence of CD8^+^ T Cells

Accumulation of CD28^null^ T cells is one of the most prominent changes during immune senescence. Similarly, the expression of CD57, a terminal differentiation marker, on T cells is considered a surrogate marker for the replicative senescence of T cells [113]. CD28^null^CD57^+^CD8^+^ T cells are recognized as activated senescent cells that produce a large amount of pro-inflammatory cytokines and highly cytotoxic molecules. The frequency of CD28^null^CD57^+^CD8^+^ T cell populations has been correlated with cardiovascular mortality six months after acute myocardial infarction [114]. Although human immunodeficiency virus (HIV)-infected individuals are at increased risk for CVD, CD28^null^CD57^+^CD^8+^ T cells were increased in HIV-infected women, and their frequency was associated with potential carotid artery disease [115]. 

It has been highly debated whether telomere shortening in patients with CAD is acquired or genetic. Persistent infection with cytomegalovirus (CMV) causes significant clonal expansion of specific T cells and telomere shortening as a result of continuous exposure to viral antigens, thus promoting T cell senescence [105,116,117]. Telomere length shortening was particularly pronounced in CD28^null^CD57^+^CD^8+^ T cells obtained from CMV-seropositive patients with CAD, but not from those without CAD; it also was correlated with a decrease in left ventricular function [118]. CMV-seropositive older adults had higher levels of effector memory T cells and effector memory T cells re-expressing CD45RA (TEMRA) in CD^4+^ and CD^8+^ T cell populations than did CMV-seronegative older adults. In addition, CMV-specific CD^8+^ T cells express CD57 [109]. TEMRA is a hallmark of cellular senescence, including reduced proliferation, defective mitochondrial function, and elevated levels of both ROS and p38. Despite this, it produces large amounts of pro-inflammatory cytokines such as IFN-γ and TNFα [115] and has the ability to exert high cytotoxic activity against CMV-infected cells [116,117,118,119,120,121]. Inhibition of p38 MAPK signaling in senescent CD8^+^ T cells restores proliferation, telomerase activity, mitochondrial biogenesis, and immune fitness [122]. These observations suggest that immune senescence of T cells due to CMV infection, which increases the inflammatory phenotype but decreases their proliferative capacity, may be an advantageous adaptation for maintaining the T cell repertoire and inhibiting reactivation of latent infection. In contrast, senescent T cells may cause systemic inflammation and contribute to the development of CVD. Since immune senescence may be an adaptive response to maintain homeostasis, the impact of age-related immune senescence on the pathogenesis of CVD and therapeutic strategies targeting immune senescence need to be carefully considered.

### 4.3. Plasma OPN Levels Predict Prognosis in Patients with Cardiovascular Diseases

T cell immunosenescence play a central role in the pathogenesis of chronic inflammation in visceral adipose tissue through the production of OPN in obese mice [34]. OPN is a matricellular protein that mediates diverse biological functions [51,52,53,123]. It functions as a proinflammatory cytokine and promotes cell-mediated immune responses [54,55]. It has been implicated in a number of CVDs, including CAD, HF, myocarditis, dilated cardiomyopathy, and atherosclerosis in experimental animal models [52,124,125,126,127,128,129]. It is also a strong predictor of adverse outcomes in patients with CVD [130,131,132].

Plasma OPN levels are increased and inversely correlated with LV ejection fraction (EF) in patients with stable CAD [132,133,134,135,136,137]. In patients with HF exhibiting reduced EF, plasma OPN levels were increased and correlated with severity as assessed by the New York Heart Association (NYHA) classification. Furthermore, plasma OPN levels predicted death within four years of follow-up in these patients [138].

Plasma OPN levels drastically changed in a time-dependent manner in patients who underwent successful reperfusion after anterior-wall acute MI, beginning to increase on day two, peaking on day three, and persisting until day 14 [139]. We previously reported that the major source of OPN is cardiac macrophages of the infarcted myocardium; this OPN directly contributes to the phagocytic clearance of dead cells and the reparative fibrotic response in wound healing in EGFP-*Spp1*-KI reporter mouse hearts subjected to MI [127,128]. Transcription of *Spp1* (the gene encoding OPN) in cardiac macrophages peaked on day 3 after myocardial infarction and completely disappeared 28 days later.

Plasma OPN levels have been associated with rapid coronary plaque progression and in-stent restenosis [140]; this increase in OPN expression in plaques is correlated with the formation of ulceration, inflammation, and unstable plaques in patients undergoing carotid endarterectomy [141]. 

Epicardial and pericardial fat are associated with cardiovascular risk [142,143,144,145]. In humans, OPN expression increases in the epicardial fat of patients with CAD compared to control subjects and is associated with the presence of calcified atherosclerotic plaques. [145] We have confirmed that senescent CD153^+^PD-1^+^CD44^hi^CD^4+^T cells markedly increased and produced a large amount of OPN in the epicardial fat of HFD-induced obese mice (K. Shirakawa, unpublished observation). These findings suggest that senescent CD^4+^ T cells in epicardial fat, at least in part, contribute to the development and worsening of CVD in obese patients.

### 4.4. Possible Mechanism of Exacerbation of CVD by OPN

Cardiac fibroblasts play a pivotal role in cardiac remodeling by modulating the extent and composition of the ECM. OPN acts on fibroblasts to induce migration and regulate the expression of multiple fibroblast-promoting genes, including alpha smooth muscle actin (α-SMA), connective tissue growth factor, and transforming growth factor-β1 (TGF-β1) [146]. Matrix metalloproteinases (MMPs) are members of the ECM protease family responsible for collagen degradation in the ECM [147]. They play an important role in tissue remodeling in heart diseases, including MI, pressure overload-induced hypertrophy, and dilated cardiomyopathy (DCM). The activation of MMP induces a reduction in cardiac tissue tensile strength, followed by systolic and diastolic dysfunction [144]. OPN causes an upregulation of tissue inhibitors of MMP and a downregulation of MMP-1 expression in cardiac fibroblasts [148]. Furthermore, OPN inhibits IL-1β-induced activation of MMPs via the activation of protein kinase C-ζ in adult rat cardiac fibroblasts, leading to enhanced collagen deposition after MI [149]. OPN also activates proinflammatory functions, including the production of IFN-γ and IL-17 in T cells and pathogenic antibodies in B cells. It suppresses anti-inflammatory functions, such as IL-10 production by regulatory T cells and B cells [150].

The supply of inflammatory leukocytes from the bone marrow niche is deeply involved in the pathogenesis of atherosclerosis and its complications [151]. Aged HSCs are more likely to enter the myeloid lineage and less likely to enter the lymphoid lineage [152]. A decrease in OPN in the bone marrow stroma contributes to these age-related HSC changes. Thrombin-cleaved OPN ameliorates the age-related HSC phenotype and restores the balance between lymphoid and myeloid cells in the peripheral blood by regulating the activity of Cdc42, a small RhoGTPase that controls HSC senescence and rejuvenation, via α9β1 integrin [153]. Obesity increases the risk of CVD. However, several epidemiological studies have shown that obesity may be protective after the onset of CVD, also known as the obesity paradox [154]. The molecular mechanisms related to this paradox are not fully understood. The increased expression of OPN associated with obesity suppresses the skewing toward the myeloid lineage with aging, which may reduce the exacerbation of CVD.

The action of OPN depends on context. It is transiently produced by mobilized macrophages after myocardial infarction for wound repair, while OPN in the bone marrow stroma is thought to play an important role in maintaining cardiovascular homeostasis. On the other hand, OPN persistently secreted from senescent CD^4+^ T cells in the visceral and ectopic fat of obese patients may contribute to pathological remodeling of the cardiovascular system. Thus, immunosenescence is considered a risk factor for cardiovascular diseases common to obesity and aging (Figure 2).

## 5. Translational Perspective

Senescent T cells may be selectively eliminated in vivo with a new type of therapeutics known as senolytics, potentially affording a new approach to treat CVD. Senolytics have been developed to induce senescent cell death by targeting apoptosis, histone modifications and chaperones [41]. Suppression of AMPK-dependent MAPK activity, which is central to TEMRA cell aging, may be a therapeutic target for T cell senescence [111]. In fact, inhibition of p38 MAPK has been shown to partially restore the proliferative capacity of TEMRA cells by inducing telomerase activity [155]. However, because TEMRA cells maintain effector function and are specific for latent viruses, their depletion may allow reactivation of the virus. The CD153 vaccine may be a promising senolytic option for preventing the accumulation of senescent T cells. Recently, Yoshida et al., demonstrated that vaccine for CD153-CpG showed a significant decrease in adipose senescent T cell accumulation and concomitant improvement in glucose tolerance and insulin resistance in n HFD-induced obese mice [83]. Vaccines that act specifically on senescent T cells may also be useful in humans.

## 6. Conclusions

Accumulating evidence demonstrates the significance of immunosenescence associated with aging and obesity in the pathogenesis of CVD. It is now known that immune senescence is not caused by an overall functional deterioration of T lymphocyte populations, as previously thought, but by an increase in the proportion of a particular T lymphocyte population that has undergone cellular senescence. Selective elimination of this particular population may help to reverse the immunosenescence associated with aging and obesity. The ability to “rejuvenate” the immune system would open up new prospects for the use of immunotherapy against CVD.

## Figures and Tables

**Figure 1 cells-10-02435-f001:**
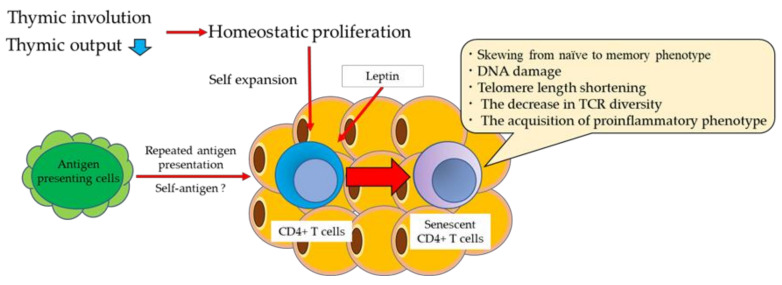
T cell senescence during obesity. Obesity leads to the skewing of the T cell subset from naïve to memory phenotypes, accelerates thymic involution, and restricts TCR diversity. Obesity-induced reduction of thymic output leads to extensive homeostatic proliferation of peripheral T cells, which may contribute to T cell senescence. VAT T cells show a senescent phenotype, including telomere length shortening, restricted TCR diversity, and production of proinflammatory cytokines.

**Figure 2 cells-10-02435-f002:**
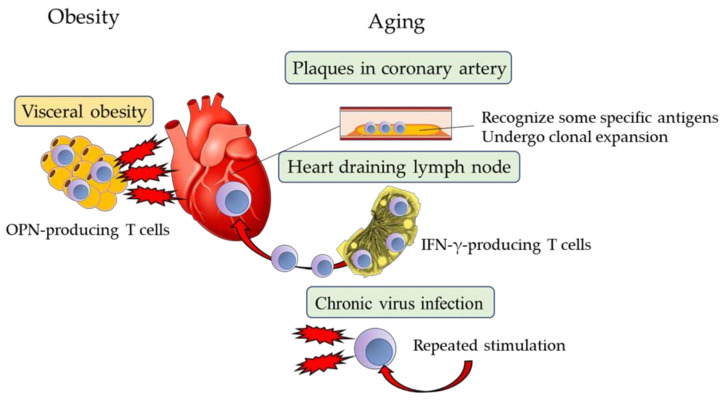
CVD pathogenesis from the perspective of immunosenescence. Proinflammatory senescent T cells recognize specific antigens, undergo clonal expansion, and infiltrate unstable coronary plaques. IFN-γ-producing T cells accumulate in the heart-draining lymph nodes of aged mice, which infiltrate the myocardium and cardiac fibrosis. Chronic viral infections, such as those by cytomegalovirus, may contribute to the development of T cell senescence. Visceral obesity accelerates T cell senescence. Senescent T cells may contribute to the pathogenesis of CVD.

## Data Availability

Not applicable.

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
