# Peer review of "T Cell Immunosenescence in Aging, Obesity, and Cardiovascular Disease"

_cells, 2021, doi:10.3390/cells10092435_

Round 1

Reviewer 1 Report

In this review, Shirakawa and Sano discuss the relationship between cardiovascular diseases and immunosenescence associated with aging and obesity.

The review is comprehensive and well written, covering the current understanding of T cell immune senescence in CVD.

Please find below my comments

-Because the manuscript is focusing mostly on T cell senescence, it would be more appropriate to change the title of the review (which is too broad) for a more “T cell” specific one. 

-Perhaps one last section addressing the therapeutic strategies that can be envisioned to revert immunosenescence of T cells in CVD or discuss the use of putative senolytic agents to rejuvenate the immune system should be developed.

Author Response

Point-by-point reply to the reviewers

Reviewer #1

In this review, Shirakawa and Sano discuss the relationship between cardiovascular diseases and immunosenescence associated with aging and obesity. The review is comprehensive and well written, covering the current understanding of T cell immune senescence in CVD. Please find below my comments.

Question 1

-Because the manuscript is focusing mostly on T cell senescence, it would be more appropriate to change the title of the review (which is too broad) for a more “T cell” specific one.

Answer 1

We changed the title to ‘T cell immunosenescence in aging, obesity, and cardiovascular disease’.

Question 2

-Perhaps one last section addressing the therapeutic strategies that can be envisioned to revert immunosenescence of T cells in CVD or discuss the use of putative senolytic agents to rejuvenate the immune system should be developed.

Answer 2

Thank you for your suggestion. We add the therapeutic strategies in the last section as follows;

“Senescent T cells may be selectively eliminated in vivo with a new type of therapeutics known as senolytics, potentially affording a new approach to treat CVD. Senolytics have been developed to induce senescent cell death by targeting apoptosis, chaperones, and histone modifications. Suppression of AMPK-dependent MAPK activity, which is central to TEMRA cell aging, may be a therapeutic target for T cell senescence. In fact, inhibition of p38 MAPK has been shown to partially restore the proliferative capacity of TEMRA cells by inducing telomerase activity. However, because TEMRA cells maintain effector function and are specific for latent viruses, their depletion may allow reactivation of the virus. The CD153 vac-cine may be a promising senolytic option for preventing the accumulation of senescent T cells. Recently, Yoshida et al., demonstrated that vaccine for CD153-CpG showed a significant decrease in adipose senescent T cell accumulation and concomitant improvement in glucose tolerance and insulin resistance in HFD-induced obese mice. Vaccines that act specifically on senescent T cells may also be useful in humans”.

We appreciate the very comprehensive and thoughtful comments and suggestions by this reviewer.

Reviewer 2 Report

The authors provide a comprehensive overview on several aspects of immune cell function in aging, obesity, and cardiovascular disease. This is a growing field of interest and a major unmet clinical need, which is why this review is likely to draw interest from a broad and interdisciplinary audience of scientists, ranging from immunologists to cardiologists. 

Specific comments:

1.) Although the manuscript is overall well written, this reviewer found it hard to read. The main reason for that might be that the authors are trying to cover too much ground. The very obvious focus of this article is on T cell senescence, and the authors should use this as a “red thread” to guide the reader through the different pathophysiologies described in aging, obesity, and cardiovascular disease. 

2.) Starting off with the title, this reviewer would favor something like “Lymphocyte immunosenescence in aging, obesity, and cardiovascular disease”. This would appear more appropriate for numerous reasons, including i) the role of myeloid cells in all three conditions is - rightfully so - not examined, ii) the roles of aging and obesity in cardiovascular disease are mainly described with a clear focus on lymphocyte immunosenescence. The abstract will have to be revised accordingly. 

3.) The authors should provide a more comprehensive introduction, which should cover all major aspects of inflammation and immune cell function, including clonal hematopoiesis and myeloid cell functions in aging, obesity and cardiovascular disease. Of note, granulocytes, monocytes and macrophages are undoubtfully central players in all three conditions. As this review has a different focus, the authors should use the introduction to guide the reader to the concept and significance of lymphocyte immunosenescence.

4.) The structure of the article is currently as follows (only main subheadings):

  1. Introduction
  2. Immunosenescence as a function of aging (not sure if “chronological” is needed here)
  3. T cell senescence in cardiovascular disease
  4. Obesity and cardiovascular disease (I would recommend not to use abbreviations in the headings)
  5. Visceral obesity and cellular senescence
  6. Obesity and T cell immunosenescence
  7. Osteopontin in cardiovascular disease
  8. Effect of weight loss on cardiovascular disease and T cell senescence
  9. Conclusion

This structure appears like a “patchwork rug” to this reviewer and made it very hard to follow the authors’ explanations. I would highly recommend to use the following structure (with appropriate subheadings):

  1. Introduction
  2. Immunosenescence as a function of aging
  3. Obesity and T cell immunosenescence
  4. Immunosenescence in cardiovascular disease
  5. Translational perspective (explained in following paragraph)
  6. Conclusions

The current paragraph on Osteopontin appears on loosely connected to lymphocyte immunosenescence/the rest of this manuscript. This should either be improved or this paragraph should be omitted in its present form. Specific information on Osteopontin could be added to "Immunosenescence in cardiovascular disease". 

5.) The authors should add a paragraph on a translational perspective. How can immunosenescence be targeted in patients? Which are the most promising experimental approaches? How would these differ from targeting inflammation in general or myeloid cell function? Exploring these points would upgrade this manuscript. 

6.) Line 254 should be “organism” instead of “organization”. 

Author Response

Point-by-point reply to the reviewers

Reviewer #2

The authors provide a comprehensive overview on several aspects of immune cell function in aging, obesity, and cardiovascular disease. This is a growing field of interest and a major unmet clinical need, which is why this review is likely to draw interest from a broad and interdisciplinary audience of scientists, ranging from immunologists to cardiologists. 

Specific comments:

Although the manuscript is overall well written, this reviewer found it hard to read. The main reason for that might be that the authors are trying to cover too much ground. The very obvious focus of this article is on T cell senescence, and the authors should use this as a “red thread” to guide the reader through the different pathophysiologies described in aging, obesity, and cardiovascular disease. 

Question 1,2

Starting off with the title, this reviewer would favor something like “Lymphocyte immunosenescence in aging, obesity, and cardiovascular disease”. This would appear more appropriate for numerous reasons, including i) the role of myeloid cells in all three conditions is - rightfully so - not examined, ii) the roles of aging and obesity in cardiovascular disease are mainly described with a clear focus on lymphocyte immunosenescence. The abstract will have to be revised accordingly. 

Answer 1, 2

We changed the title to ‘T cell immunosenescence in aging, obesity, and cardiovascular disease’ and revised the abstract.

Abstract: Although advances in preventive medicine have greatly improved prognosis, cardio-vascular disease (CVD) remains the leading cause of death worldwide. This clearly indicates that there remain residual cardiovascular risks that have not been targeted by conventional therapies. The results of multiple animal studies and clinical trials clearly indicate that inflammation is the most important residual risk and a potential target for CVD prevention. The immune cell network is intricately regulated to maintain homeostasis. Ageing- associated changes to the immune system occurs in both innate and adaptive immune cells, however T cells are most susceptible to this process. T-cell changes due to thymic degeneration and homeostatic proliferation, metabolic abnormalities, telomere length shortening, and epigenetic changes associated with aging and obesity may not only reduce normal immune function, but also induce inflammatory tendencies, a process referred to as immunosenescence.  Since the disruption of biological homeostasis by T cell immunosenescence is closely related to the development and progression of CVD via inflammation, senescent T cells is attracting attention as a new therapeutic target. In this review, we discuss the relationship between CVD and T cell immunosenescence associated with aging and obesity.

Question 3

The authors should provide a more comprehensive introduction, which should cover all major aspects of inflammation and immune cell function, including clonal hematopoiesis and myeloid cell functions in aging, obesity and cardiovascular disease. Of note, granulocytes, monocytes and macrophages are undoubtfully central players in all three conditions. As this review has a different focus, the authors should use the introduction to guide the reader to the concept and significance of lymphocyte immunosenescence.

Answer 3

This review is mainly focused on T cell senescence, so we changed the title to ‘T cell immunosenescence in aging, obesity, and cardiovascular disease’. As this reviewer pointed out, age-related changes occur in most types of cells in both innate and adaptive immune systems, however generation and homeostasis of T cells are susceptible to ageing due to the involution of the thymus. We revised the introduction, considering this point.

There is no doubt that myeloid cell-mediated inflammation associated with changes in hematopoietic function in bone marrow due to CHIP and environmental stress plays a central role when considering inflammation as a residual risk of CVD, as the reviewer mentioned.There is no doubt that myeloid cell-mediated inflammation plays a central role in the changes in hematopoietic function in bone marrow associated with CHIP and environmental stress. Based on these facts, this article describes the relationship between cellular senescence, especially T-cell senescence, and CVD risk. The following text has been inserted in the introduction to avoid misunderstanding by readers.

“It has been reported that a certain percentage of hematopoietic stem/progenitor cells with a specific genetic mutation called clonal hematopoiesis of indeterminate potential (CHIP) appear in the elderly, which is not only associated with the risk of hematopoietic tumors but also with the development of atherosclerotic cardiovascular disease (N Engl J Med. 2017;377:111–121). In addition, environmental stresses such as lack of exercise and sleep disturbance have been shown to accelerate the progression of atherosclerosis by increasing the production of inflammatory monocytes and neutrophils in the bone marrow (Nature 2019. 566:383–387) (Nat. Med. 2019. 25: 1761–1771)”.

Question 4

The structure of the article is currently as follows (only main subheadings):

  1. Introduction
  2. Immunosenescence as a function of aging (not sure if “chronological” is needed here)
  3. T cell senescence in cardiovascular disease
  4. Obesity and cardiovascular disease (I would recommend not to use abbreviations in the headings)
  5. Visceral obesity and cellular senescence
  6. Obesity and T cell immunosenescence
  7. Osteopontin in cardiovascular disease
  8. Effect of weight loss on cardiovascular disease and T cell senescence
  9. Conclusion

This structure appears like a “patchwork rug” to this reviewer and made it very hard to follow the authors’ explanations. I would highly recommend to use the following structure (with appropriate subheadings):

  1. Introduction
  2. Immunosenescence as a function of aging
  3. Obesity and T cell immunosenescence
  4. Immunosenescence in cardiovascular disease
  5. Translational perspective (explained in following paragraph)
  6. Conclusions

The current paragraph on Osteopontin appears on loosely connected to lymphocyte immunosenescence/the rest of this manuscript. This should either be improved or this paragraph should be omitted in its present form. Specific information on Osteopontin could be added to "Immunosenescence in cardiovascular disease". 

Answer 4

We have changed the structure as you pointed out. The review on Osteopontin has been moved to the section on Immunosenescence in cardiovascular disease. Thank you for your suggestion.

Question 5

The authors should add a paragraph on a translational perspective. How can immunosenescence be targeted in patients? Which are the most promising experimental approaches? How would these differ from targeting inflammation in general or myeloid cell function? Exploring these points would upgrade this manuscript. 

Answer 5

Thank you for your suggestion.

We added the therapeutic strategies in the last section as follows;

“Senescent T cells may be selectively eliminated in vivo with a new type of therapeutics known as senolytics, potentially affording a new approach to treat CVD. Senolytics have been developed to induce senescent cell death by targeting apoptosis, chaperones, and histone modifications. Suppression of AMPK-dependent MAPK activity, which is central to TEMRA cell aging, may be a therapeutic target for T cell senescence. In fact, inhibition of p38 MAPK has been shown to partially restore the proliferative capacity of TEMRA cells by inducing telomerase activity. However, because TEMRA cells maintain effector function and are specific for latent viruses, their depletion may allow reactivation of the virus. The CD153 vac-cine may be a promising senolytic option for preventing the accumulation of senescent T cells. Recently, Yoshida et al., demonstrated that vaccine for CD153-CpG showed a significant decrease in adipose senescent T cell accumulation and concomitant improvement in glucose tolerance and insulin resistance in n HFD-induced obese mice. Vaccines that act specifically on senescent T cells may also be useful in humans”.

Question 6

Line 254 should be “organism” instead of “organization”. 

Answer 6

We have fixed the mistake. Thank you for your advice.

We appreciate the very comprehensive and thoughtful comments and suggestions by this reviewer.